# Evaluation of Nitrogen and Cropping System Management in Continuous Winter Wheat Forage Production Systems

Bronc Finch [1], Joao Luis Bigatao Souza [2], Vaughn Reed [3], Raedan Sharry [2], Michaela Smith [2] and Daryl Brian Arnall [2,*]

1 Department of Crop, Soil, and Environmental Sciences, University of Arkansas, Little Rock, AR 72204, USA
2 Department of Plant and Soil Sciences, Oklahoma State University, Stillwater, OK 74078, USA
3 Department of Plant and Soil Sciences, Mississippi State University, Mississippi State, MS 39762, USA
* Correspondence: b.arnall@okstate.edu

**Abstract:** In the central Great Plains, winter wheat is used for over-winter grazing for cattle and sheep until the late spring months, when livestock are moved to grass pasture. As the popularity of summer cover crops increases, interest in their use in forage production systems increases as well. There is specific interest in the opportunity to increase productivity by the inclusion of a crop grown in the fallow season of winter wheat fields. The intensification of systems in a resource (water and/or nitrogen) limited region could decrease winter wheat forage production influencing a system's ability to sustain continuous forage production. Nitrogen (N) management could be effective in mitigating negative impacts on winter wheat. The objective of this study is to evaluate the influence of different summer forage crop species and different N management strategies in a multi-year continuous winter wheat forage production system in the central Grain Plains. Increased production of dry matter and crude protein was observed by implementing summer forage crops into a winter wheat forage system. A deleterious effect of summer crops compared to traditional fallow periods was observed but mitigated by the split application of N even compared to the same rate applied at pre-plant.

**Keywords:** cover crops; continuous forage; graze-out; nitrogen; split application; crude protein; gain

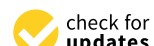



## 1. Introduction

Continuous feed production by growing both a winter forage and summer forage on the same field is an opportunity for livestock producers to increase the productivity and profitability of their operation. Traditionally, fields where winter wheat (*Triticum aestivum* L.) is grown and harvested/grazed for forage are then left to fallow or left without establishing other forage crops during the summer. This fallow season is used to increase the chances of successful establishment and growth of winter wheat in variable, unpredictable, dryland growing regions [1]. Producers may look to further increase the productivity of their operation by intensifying the system, with the inclusion of a crop grown in the general fallow season of their fields.

Intensification of cropping systems by replacing summer fallow periods has shown a mixture of results regarding impacts on soil moisture storage, subsequent crop production, and soil organic carbon stores, among others [1–7]. Of these, some consistent results are found, such as reduced soil erosion, reduced nitrate (NO₃–N) leaching, and increased soil NO₃–N accumulation following summer legume crops [8–11]. However, a negative impact on the production of subsequent crops has been reported following summer forage crop production [3,5]. In their evaluation of cool-season forage cover crops on teff grass production, Baxter et al. [12] reported minimal impact on the subsequent teff forage accumulation by the use of cool-season forage cover crops. The authors further noted that the use of winter wheat decreased soil volumetric water production greater than the other cool-season forage crops. This decrease in soil water by winter wheat is an important consideration

when evaluating the use of summer forage cover crops in a continuous winter wheat forage production system.

Nitrogen (N) management can be used to increase the productivity of forage crops and continuous forage production systems. Most N management studies have shown a linear increase in biomass yield with incremental increases in N applications [13–17]. Similarly, summer forage crops have been reported to increase biomass production with increased N applications [18,19]. Khalil et al. [15], which used higher N rates than most other studies (120 kg N ha$^{-1}$), reported a plateau at which additional N application did not increase biomass production.

The N management for the entire system can also impact the forage productivity. For most of the central Great Plains, N applied for winter wheat forage production is applied at planting, which is represented in most studies. However, Thomason et al. [20] observed an increase in nitrogen use efficiency (NUE) when N was applied in the split application, and Naveed et al. [21] observed greater biomass production. Increased NUE of a forage crop could not only decrease the risk of wasting applied N and increase profitability but also decrease the risk of N environmental pollution [22]. Kanampiu et al. [23] found that when winter wheat pre-plant N rate was increased, grain and forage biomass yield was also increased but at the cost of increasing N losses as well. Split application of N could be a tool for increasing biomass production while reducing the chances of losing N, similar to the findings of Thomas et al. [20] and Naveed et al. [21]. When looking at the summer forage system, the use of legume crops for fallow period replacement can increase the available soil $NO_3$–N levels for subsequent crops but could also reduce the $NO_3$ returned to the soil compared to when used for cover cropping [9,10].

Although previous studies have reported the effects of using summer forage cover crops in winter wheat systems, as well as the influence of N management on forage productivity, few have evaluated the synergistic effects of summer forage cropping and N management in a continuous forage production system. Therefore, the objective of this study is to evaluate the influence of different summer forage crop species and different N management strategies in a multi-year continuous winter wheat forage production system in central Oklahoma. Results from this study could provide producers with information that could increase the productivity and profitability of their forage cropping systems.

## 2. Materials and Methods

### 2.1. Field Study

This study was conducted over three years, from 2018 to 2021, totaling three winter wheat seasons and two summer seasons. The experiment was conducted in two locations, South Central Research and Extension Center (SCREC) in Chickasha, Oklahoma (35° 2′ 18.45″ N, 97° 54′ 37.81″ W) on a McLain silt clay loam soil, and Lake Carl Blackwell Research Farm (LCB), near Stillwater, Oklahoma (36° 9′ 4.97″ N, 97° 17′ 23.92″ W) on a Pulaski fine sandy loam soil. The experiment was established as a split-plot randomized complete block design with a three-by-four-by-two factorial (Table 1), with four replications. The primary treatment factor was winter wheat N management with three levels: 67 kg N ha$^{-1}$ (low) at pre-plant, 135 kg N ha$^{-1}$ (high) at pre-plant, and a split application of 67 kg N ha$^{-1}$ applied at pre-plant with subsequent top-dress application of 67 kg N ha$^{-1}$ applied at the first winter wheat harvest or spring green-up, whichever event happened later, as urea (($NH_2$)$_2$CO, 46% N). The secondary treatment factor was summer cropping with four levels: summer fallow, monoculture cowpea (*Vigna unguiculata* L.) (Iron & Clay–MBS Seed Inc. Denton, TX, USA) planted at 67 kg seed ha$^{-1}$, monoculture pearl millet (*Pennisetum glaucum* L.) (Tifleaf 3 Hybrid Pearl Millet–Hancock Seed & Co.; Dade City, FL, USA) planted at 22 kg seed ha$^{-1}$, a cowpea pearl millet mixture (75% cowpea and 25% pearl millet, by weight) planted at 34 kg cowpea ha$^{-1}$ and 11 kg pearl millet ha$^{-1}$ within each of the primary factors. Pearl millet was outcompeted by a flush of crabgrass (*Digitaria sanguinalis* L.) in both summer seasons at LCB; therefore, the crabgrass was used as the grass species for that location. Due to extended dry periods and hot

temperatures in both summer seasons at SCREC, no summer crop germination occurred in the 2019 summer, and pearl millet did not germinate in the 2020 summer. This resulted in the mixture plots having cowpeas germinated at a planting rate of 34 kg ha$^{-1}$, denoted as a half-planted rate of cowpeas. Within each of the secondary factors, the tertiary factor was summer N with two levels of N application, 0 or 34 kg N ha$^{-1}$ applied as liquid urea-ammonium nitrate (UAN,28% N).

**Table 1.** Continuous winter wheat forage treatment structures for Lake Carl Blackwell (Left) and SCREC (Right). Winter wheat N rate at pre-plant and top-dress timings in kg N ha$^{-1}$, summer forage crop species, and summer N application rate in kg ha$^{-1}$.

| Lake Carl Blackwell | | | SCREC | | |
|---|---|---|---|---|---|
| **Winter Wheat** | **Summer Forage** | | **Winter Wheat** | **Summer Forage** | |
| **Nitrogen** | **Crop** | **N (kg ha$^{-1}$)** | **Nitrogen** | **Crop** | **N (kg ha$^{-1}$)** |
| Low (67 kg ha$^{-1}$) | Fallow | 0 30 | Low (67 kg ha$^{-1}$) | Fallow | 0 30 |
| | Crabgrass | 0 30 | | Cowpea | 0 30 |
| | Cowpea | 0 30 | | Cowpea 0.5× | 0 30 |
| | Crabgrass & Cowpea | 0 30 | High (135 kg ha$^{-1}$) | Fallow | 0 30 |
| High (135 kg ha$^{-1}$) | Fallow | 0 30 | | Cowpea | 0 30 |
| | Crabgrass | 0 30 | | Cowpea 0.5× | 0 30 |
| | Cowpea | 0 30 | Split (67 kg ha$^{-1}$ pre-plant) (67 kg ha$^{-1}$ top dress) | Fallow | 0 30 |
| | Crabgrass & Cowpea | 0 30 | | Cowpea | 0 30 |
| Split (67 kg ha$^{-1}$ pre-plant) (67 kg ha$^{-1}$ top dress) | Fallow | 0 30 | | Cowpea 0.5× | 0 30 |
| | Crabgrass | 0 30 | | | |
| | Cowpea | 0 30 | | | |
| | Crabgrass & Cowpea | 0 30 | | | |

Winter wheat was planted using Gallagher variety (Oklahoma Genetics Inc.; Stillwater, OK, USA) at 145 kg ha$^{-1}$ using a John Deere 1590 no-till drill (John Deere Mfg.; Moline, IL, USA) at SCREC, and 135 kg ha$^{-1}$ using a Great Plains 1006NT no-till drill (Great Plains Mfg.; Salina, KS, USA) at LCB. Summer crops were planted using a Great Plains 3P506NT no-till drill (Great Plains Mfg.; Salina, KS, USA) at both locations. Field management was conducted to reflect traditional rainfed winter wheat forage production methods, including pesticide management.

### 2.2. Soil Analysis

Prior to each winter wheat season, pre-plant 0–15 cm composite soil samples (Fifteen 2.5 cm diameter cores per composite) were collected from each subplot (Table 2). These samples were also collected after the trial was complete. Samples were analyzed for soil inorganic nitrogen of nitrate ($NO_3$–N) and ammonium ($NH_4$–N) concentrations by Flow Injection Autoanalyzer (LACHAT, 1994—QuickChem Method 12-107-04-1-B—LACHAT Instrument, Milwaukee, WI, USA) using a 1 M KCl extraction method. Soil total carbon (Total

C) and total nitrogen (Total N) were determined using a dry combustion carbon/nitrogen analyzer (CN 628, LECO Corporation, St. Joseph, MI, USA). Soil pH was determined by pH electrode measurement of a 1:1 soil: water solution after a 30 min equilibration period. Phosphorus (P) and potassium (K) were extracted with Mehlich 3 extraction solution [24] and analyzed using inductively coupled plasma-optical emission spectrometry [ICP-OES; SPECTRO Analytical Instruments; GmbH, Kleve, Germany].

**Table 2.** Mean, minimum, and maximum soil chemical properties at trial initiation of each location of the study. Lake Carl Blackwell (LCB) and the South-Central Research and Extension Station (SCREC).

| Location | | pH | Inorganic N | M3P | K | Total N | Total C |
|---|---|---|---|---|---|---|---|
| | | | (kg ha$^{-1}$) | (ppm) | | (g kg$^{-1}$) | |
| LCB | Min | 5.6 | 7 | 64 | 822 | 0.70 | 6.20 |
| | Mean | 5.9 | 13 | 91 | 457 | 0.90 | 7.50 |
| | Max | 6.1 | 19 | 130 | 380 | 1.00 | 9.10 |
| SCREC | Min | 5.9 | 33 | 54 | 686 | 1.30 | 12.20 |
| | Mean | 6.2 | 57 | 90 | 836 | 1.50 | 14.50 |
| | Max | 7.1 | 98 | 134 | 1016 | 1.70 | 17.20 |

### 2.3. Biomass Harvest

Biomass harvest of each subplot was accomplished using a flail-type forage harvester (Carter Mfg. Co.; Brookston, IN, USA) by collecting the weight of all biomass greater than a height of 5 cm from a 1 m × 6 m area. Sub-samples were collected to analyze for moisture and quality analysis and were dried for 24 h in a forced air dryer at 65 °C prior to grinding to pass a 1 mm sieve. Forage yield was reported as Mg ha$^{-1}$ dry matter (DM), calculated from wet weight in the field and using the percent moisture derived from the sub-samples to determine DM. Forage quality analysis of crude protein was determined using a dry combustion carbon/nitrogen analyzer (CN 628, LECO Corporation, St. Joseph, MI, USA). Crude protein yield (CP) was calculated by multiplying the crude protein concentration by the total dry matter biomass produced.

### 2.4. Statistical Analysis

Statistical analysis was conducted using PROC GLM in SAS software, version 9.4 (SAS Institute Inc., Cary, NC, USA). Dry matter, CP, and GY were analyzed for the interaction of treatment factors and year within cropping season within the location, where locations were kept separate due to a difference in treatments caused by environmental factors. Non-significant year interactions were pooled and analyzed for overall treatment influence. Mean separation was conducted using a Fisher's t-test for least significant difference (LSD) analysis at an alpha of 0.05.

## 3. Results

### 3.1. Dry Matter Biomass Yield

Winter wheat DM yield across the entire system was influenced by the interaction of year and winter wheat N application at both locations ($p \leq 0.0099$), the interaction of year and summer crop at LCB ($p = 0.0359$), and the main effect of summer crop at SCREC ($p = 0.0266$) (Table 3). The 2018–2019 winter wheat DM was increased by the split application to 15 Mg ha$^{-1}$ at LCB ($p < 0.0001$) and 14 Mg ha$^{-1}$ at SCREC ($p = 0.0061$) (Figure 1), compared to 12 Mg ha$^{-1}$ at both locations when the same N rate was applied pre-plant. The pre-plant applications yielded 12 Mg ha$^{-1}$ DM when a high N rate was applied and 9 Mg ha$^{-1}$ when a low N rate was applied at LCB. The pre-plant applications were not different at SCREC and yielded an average of 12 Mg ha$^{-1}$ DM in 2018–2019.

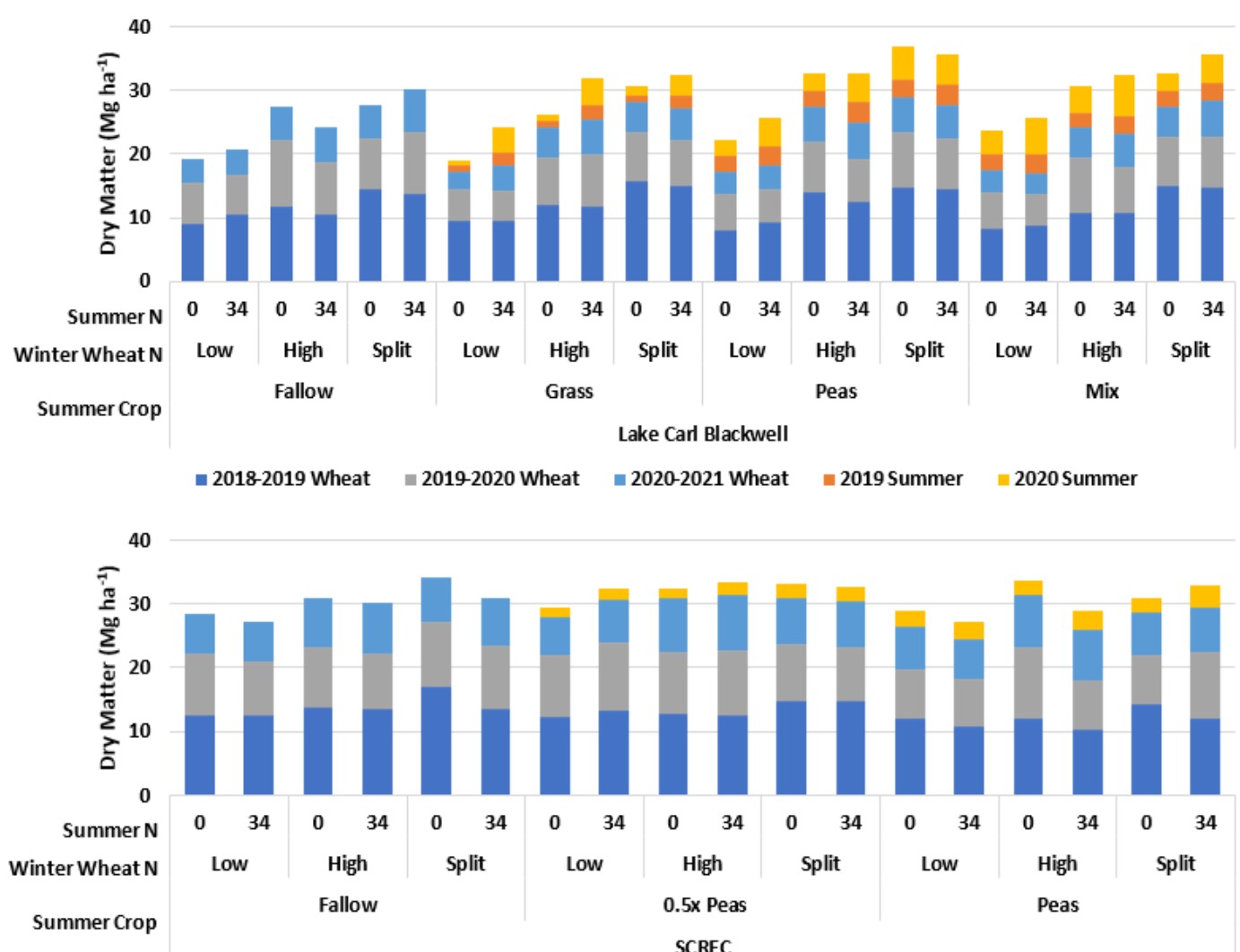

**Figure 1.** Average dry matter biomass production (Mg ha$^{-1}$) from each season of production for LCB (Top) and SCREC (Bottom). Means are presented for each of the summer N application rates (kg N ha$^{-1}$) within each of the summer crop species: fallow, cowpeas (Peas), crabgrass (Grass), cowpea-crabgrass mixture (Mix), and half-planted rate of Cowpeas (0.5× Peas), within each of the winter wheat N applications Low (67 kg N ha$^{-1}$ pre-plant), High (135 kg N ha$^{-1}$ pre-plant), and Split (67 kg N ha$^{-1}$ pre-plant and 67 kg N ha$^{-1}$ top-dress).

**Table 3.** ANOVA table with degrees of freedom (DF), sums of squares, mean squares, *F*-value, and *p*-value for each of the sources of variances for winter wheat dry matter production at two locations in Oklahoma.

| | Summer Crop Dry Matter Biomass | | | | | | | | | |
|---|---|---|---|---|---|---|---|---|---|---|
| | SCREC | | | | | LCB | | | | |
| Source | DF | Sum of Squares | Mean Square | F-Value | *p*-Value | DF | Sum of Squares | Mean Square | F-Value | *p*-Value |
| Model | 56 | 2284.40 | 40.79 | 7.44 | <0.0001 ** | 74 | 3441.95 | 46.51 | 18.40 | <0.0001 ** |
| Winter Wheat Nitrogen (WN) | 2 | 57.12 | 28.56 | 5.21 | 0.0061 ** | 2 | 554.00 | 277.00 | 109.57 | <0.0001 ** |
| Summer Crop (SC) | 2 | 40.41 | 20.20 | 3.68 | 0.0266 * | 3 | 17.49 | 5.83 | 2.31 | 0.0778 |
| Summer Nitrogen (SN) | 1 | 6.80 | 6.80 | 1.24 | 0.2666 | 1 | 0.17 | 0.17 | 0.07 | 0.7980 |

**Table 3.** *Cont.*

| | | | Summer Crop Dry Matter Biomass | | | | | | | |
| | | SCREC | | | | LCB | | | | |
| Source | DF | Sum of Squares | Mean Square | F-Value | *p*-Value | DF | Sum of Squares | Mean Square | F-Value | *p*-Value |
|---|---|---|---|---|---|---|---|---|---|---|
| Year (YR) | 2 | 1522.88 | 761.44 | 138.82 | <0.0001 ** | 2 | 2541.22 | 1270.61 | 502.59 | <0.0001 ** |
| WN × SC | 4 | 13.04 | 3.26 | 0.59 | 0.6671 | 6 | 7.64 | 1.27 | 0.50 | 0.8053 |
| WN × SN | 2 | 3.94 | 1.97 | 0.36 | 0.6988 | 2 | 7.01 | 3.50 | 1.39 | 0.2524 |
| SC × SN | 2 | 11.92 | 5.96 | 1.09 | 0.3391 | 3 | 1.67 | 0.56 | 0.22 | 0.8825 |
| YR × WN | 4 | 74.72 | 18.68 | 3.41 | 0.0099 ** | 4 | 155.24 | 38.81 | 15.35 | <0.0001 ** |
| YR × SC | 4 | 29.61 | 7.40 | 1.35 | 0.2523 | 6 | 34.88 | 5.81 | 2.30 | 0.0359 * |
| YR × SN | 2 | 14.37 | 7.18 | 1.31 | 0.2719 | 2 | 9.16 | 4.58 | 1.81 | 0.1659 |
| WN × SC × SN | 4 | 19.64 | 4.91 | 0.9 | 0.4675 | 6 | 13.63 | 2.27 | 0.90 | 0.4969 |
| YR × WN × SC | 8 | 27.45 | 3.43 | 0.63 | 0.7560 | 12 | 32.28 | 2.69 | 1.06 | 0.3920 |
| YR × WN × SN | 4 | 22.29 | 5.57 | 1.02 | 0.3999 | 4 | 9.07 | 2.27 | 0.9 | 0.4665 |
| YR × SC × SN | 4 | 6.56 | 1.64 | 0.3 | 0.8784 | 6 | 2.92 | 0.49 | 0.19 | 0.9787 |
| YR × WN × SC × SN | 8 | 28.78 | 3.60 | 0.66 | 0.7300 | 12 | 12.53 | 1.04 | 0.41 | 0.9575 |

* *p*-value significant at 95% level; ** *p*-value significant at 99% level

Winter wheat DM production in 2019–2020 did not respond to treatments at the SCREC location ($p = 0.3555$). At the LCB location, the application of a high N rate, regardless of timing, increased DM biomass yield to 8.1 Mg ha$^{-1}$ on average, compared to 5.5 Mg ha$^{-1}$ produced by pre-plant applications of a low N rate (Figure 1; Table A1). The 2020–2021 winter wheat season at LCB yielded the highest DM biomass production at 5.3 Mg ha$^{-1}$ when a high N rate was applied, regardless of timing, compared to 3.6 Mg ha$^{-1}$ when only a low N rate was applied pre-plant for winter wheat. The application of a high N rate at SCREC increased winter wheat DM production to 8.2 Mg ha$^{-1}$, compared to the split application of winter wheat N yielding 7.1 Mg ha$^{-1}$ and the pre-plant low N rate yielding 6.9 Mg ha$^{-1}$.

The use of a summer fallow replacement forage crop at LCB in 2019 reduced the 2019–2020 winter wheat DM production to 6.9 Mg ha$^{-1}$ on average, in comparison to 8.2 Mg ha$^{-1}$ produced by wheat following a summer fallow period. Winter wheat DM production in 2020–2021 was not influenced by the previous summer cropping treatments. Summer crop establishment in 2019 was unsuccessful at the SCREC, as mentioned earlier, due to extended dry periods following planting, while in the summer of 2020, only cowpeas were established. The use of full-rate cowpeas decreased the total winter wheat DM biomass production at SCREC by 1.8 Mg ha$^{-1}$ on average compared to summer fallow. No other treatments influenced winter wheat DM production during the term of the study.

Summer crop DM biomass production was influenced by the interaction of winter N application and summer crop species ($p = 0.0247$), the interaction of summer crop species and summer N application ($p = 0.0467$), and the interaction of year by summer N application ($p = 0.0006$) at the LCB location (Table 4). No treatment differences were observed in the summer seasons at SCREC ($p = 0.1689$). Monoculture crabgrass following a low N rate and split winter wheat N applications yielded lower DM production (1.9 Mg ha$^{-1}$) than all other combinations except monoculture crabgrass following a high N rate pre-plant winter wheat N application (2.1 Mg ha$^{-1}$). The use of monoculture cowpeas following a high N rate winter wheat N application, regardless of timing, and a cowpea crabgrass mixture following pre-plant winter wheat N application, regardless of rate, yielded an average of 3.7 Mg ha$^{-1}$ DM higher than all treatments. With the exception of cowpeas following a low N rate pre-plant winter wheat application and cowpea crabgrass mixture following a split application of winter wheat N. When monoculture crabgrass was left unfertilized during the summer, the lowest DM yield of 1.0 Mg ha$^{-1}$ was produced. The application of 34 kg N ha$^{-1}$ to a monoculture or mixed cowpea summer forage produced the greatest DM production with an average of 4.0 Mg ha$^{-1}$. In 2019, LCB had increased

DM production from 2.0 Mg ha$^{-1}$ to 2.7 Mg ha$^{-1}$ with the application of 34 kg N ha$^{-1}$ compared to without. Similarly, the 34 kg N ha$^{-1}$ applied to summer crops in 2020 at LCB increased DM production from 2.7 Mg ha$^{-1}$ to 4.6 Mg ha$^{-1}$ on average.

**Table 4.** ANOVA table with degrees of freedom (DF), sums of squares, mean squares, *F*-value, and *p*-value for each of the sources of variances for summer crop dry matter production at two locations in Oklahoma.

| | | | | | | | | | | |
|---|---|---|---|---|---|---|---|---|---|---|
| | | | **Summer Crop Dry Matter Biomass** | | | | | | | |
| | | **SCREC** | | | | | **LCB** | | | |
| **Source** | **DF** | **Sum of Squares** | **Mean Square** | **F-Value** | ***p*-Value** | **DF** | **Sum of Squares** | **Mean Square** | **F-Value** | ***p*-Value** |
| Model | 14 | 15.40 | 1.10 | 1.54 | 0.1689 | 38 | 272.96 | 7.18 | 6.37 | <0.0001 ** |
| Winter Wheat Nitrogen (WN) | 2 | 1.05 | 0.52 | 0.73 | 0.4906 | 2 | 1.06 | 0.53 | 0.47 | 0.6249 |
| Summer Crop (SC) | 1 | 6.31 | 6.31 | 8.82 | 0.0065 | 2 | 75.27 | 37.64 | 33.39 | <0.0001 ** |
| Summer Nitrogen (SN) | 1 | 1.76 | 1.76 | 2.47 | 0.1290 | 1 | 60.53 | 60.53 | 53.7 | <0.0001 ** |
| Year (YR) | - | - | - | - | - | 1 | 60.74 | 60.74 | 53.88 | <0.0001 ** |
| WN × SC | 2 | 0.90 | 0.45 | 0.63 | 0.5430 | 4 | 13.16 | 3.29 | 2.92 | 0.0247 * |
| WN × SN | 2 | 0.61 | 0.31 | 0.43 | 0.6567 | 2 | 4.37 | 2.18 | 1.94 | 0.1491 |
| SC × SN | 1 | 0.31 | 0.31 | 0.44 | 0.5148 | 2 | 7.11 | 3.56 | 3.16 | 0.0467 * |
| YR × WN | - | - | - | - | - | 2 | 1.40 | 0.70 | 0.62 | 0.5388 |
| YR × SC | - | - | - | - | - | 2 | 5.58 | 2.79 | 2.48 | 0.0889 |
| YR × SN | - | - | - | - | - | 1 | 14.21 | 14.21 | 12.61 | 0.0006 ** |
| WN × SC × SN | 2 | 0.76 | 0.38 | 0.53 | 0.5956 | 4 | 1.50 | 0.38 | 0.33 | 0.8547 |
| YR × WN × SC | - | - | - | - | - | 4 | 10.12 | 2.53 | 2.24 | 0.0692 |
| YR × WN × SN | - | - | - | - | - | 2 | 2.18 | 1.09 | 0.97 | 0.3836 |
| YR × SC × SN | - | - | - | - | - | 2 | 1.78 | 0.89 | 0.79 | 0.4568 |
| YR × WN × SC × SN | - | - | - | - | - | 4 | 2.59 | 0.65 | 0.57 | 0.6820 |

* *p*-value significant at 95% level; ** *p*-value significant at 99% level

### 3.2. Crude Protein Yield

Winter wheat crude protein yield (CP) of the total system was influenced by the main effect of summer crop ($p = 0.0117$) species at SCREC, the interaction between winter wheat and summer N applications ($p = 0.0287$), the interaction of year and summer crop species ($p < 0.0001$) at LCB, and the interaction of year and winter N applications at both locations ($p \leq 0.0132$) (Table 5). The use of a fully planted rate of cowpeas for summer forage decreased the winter wheat CP yield by 0.4 Mg ha$^{-1}$ compared to the fallow and half-planted rate of cowpeas, which produced an average of 3.5 Mg ha$^{-1}$ CP yield at SCREC (Figure 2; Table A2). The split application of winter wheat N at LCB yielded the highest CP production regardless of summer N application, with an average CP yield of 0.86 Mg ha$^{-1}$. The application of a low pre-plant winter wheat N rate produced the lowest CP yield regardless of summer N application at 0.49 Mg ha$^{-1}$. When no summer N was applied, the application timing of the high N rate was not influential on the CP yield, with an average yield of 0.81 Mg ha$^{-1}$ at LCB.

**Table 5.** ANOVA table with degrees of freedom (DF), sums of squares, mean squares, *F*-value, and *p*-value for each of the sources of variances for winter wheat crude protein yield at two locations in Oklahoma.

| | | | | | | | | | | |
|---|---|---|---|---|---|---|---|---|---|---|
| | | | **Winter Wheat Crude Protein Yield** | | | | | | | |
| | | **SCREC** | | | | | **LCB** | | | |
| **Source** | **DF** | **Sum of Squares** | **Mean Square** | **F-Value** | ***p*-Value** | **DF** | **Sum of Squares** | **Mean Square** | **F-Value** | ***p*-Value** |
| Model | 56 | 36.85 | 0.66 | 6.88 | <0.0001 ** | 74 | 17.98 | 0.24 | 9.01 | <0.0001 ** |

**Table 5.** *Cont.*

| | | SCREC | | | | | LCB | | | |
|---|---|---|---|---|---|---|---|---|---|---|
| | | | | | **Winter Wheat Crude Protein Yield** | | | | | |
| **Source** | **DF** | **Sum of Squares** | **Mean Square** | **F-Value** | ***p*-Value** | **DF** | **Sum of Squares** | **Mean Square** | **F-Value** | ***p*-Value** |
| Winter Wheat Nitrogen (WN) | 2 | 1.75 | 0.88 | 9.17 | 0.0001 ** | 2 | 6.67 | 3.34 | 123.7 | <0.0001 ** |
| Summer Crop (SC) | 2 | 0.87 | 0.43 | 4.54 | 0.0117 * | 3 | 0.26 | 0.09 | 3.20 | 0.0243 * |
| Summer Nitrogen (SN) | 1 | 0.05 | 0.05 | 0.52 | 0.4725 | 1 | 0.001 | 0.001 | 0.03 | 0.8554 |
| Year (YR) | 2 | 23.79 | 11.90 | 124.41 | <0.0001 ** | 2 | 7.93 | 3.97 | 147.05 | <0.0001 ** |
| WN × SC | 4 | 0.25 | 0.06 | 0.65 | 0.6282 | 6 | 0.09 | 0.02 | 0.57 | 0.7518 |
| WN × SN | 2 | 0.03 | 0.02 | 0.16 | 0.8510 | 2 | 0.19 | 0.10 | 3.61 | 0.0287 * |
| SC × SN | 2 | 0.19 | 0.10 | 0.99 | 0.3715 | 3 | 0.02 | 0.01 | 0.27 | 0.8463 |
| YR × WN | 4 | 1.24 | 0.31 | 3.23 | 0.0132 * | 4 | 0.86 | 0.22 | 8.01 | <0.0001 ** |
| YR × SC | 4 | 0.31 | 0.08 | 0.80 | 0.5240 | 6 | 0.62 | 0.10 | 3.85 | 0.0011 ** |
| YR × SN | 2 | 0.20 | 0.10 | 1.02 | 0.3621 | 2 | 0.12 | 0.06 | 2.19 | 0.1149 |
| WN × SC × SN | 4 | 0.39 | 0.10 | 1.02 | 0.3996 | 6 | 0.14 | 0.02 | 0.89 | 0.5050 |
| YR × WN × SC | 8 | 0.53 | 0.07 | 0.70 | 0.6942 | 12 | 0.41 | 0.03 | 1.26 | 0.2413 |
| YR × WN × SN | 4 | 0.17 | 0.04 | 0.45 | 0.7719 | 4 | 0.09 | 0.02 | 0.80 | 0.5289 |
| YR × SC × SN | 4 | 0.24 | 0.06 | 0.62 | 0.6476 | 6 | 0.02 | 0.00 | 0.12 | 0.9941 |
| YR × WN × SC × SN | 8 | 0.45 | 0.06 | 0.59 | 0.7890 | 12 | 0.33 | 0.03 | 1.01 | 0.4397 |

* *p*-value significant at 95% level; ** *p*-value significant at 99% level.

A year and summer crop species interaction were only significant for the 2019–2020 winter wheat CP production, where winter wheat CP yield was decreased from 0.97 Mg ha$^{-1}$ when wheat followed a fallow summer to 0.77 Mg ha$^{-1}$ when wheat followed any summer crop. The year-by-winter wheat N application interaction revealed the split application produced higher CP than either of the pre-plant applications by at least 0.23 and 0.25 Mg ha$^{-1}$ at SCREC (1.7 Mg ha$^{-1}$) and LCB (1.1 Mg ha$^{-1}$), respectively, in the 2018–2019 winter wheat season. Moreover, in the 2018–2019 winter wheat season at LCB, the high N pre-plant rate increased CP yield by 0.24 Mg ha$^{-1}$ compared to the low pre-plant N application rate (0.58 Mg ha$^{-1}$ CP). In the 2019–2020 winter wheat season, the application of a high N rate, regardless of timing, yielded an average of 0.94 Mg ha$^{-1}$ CP, higher than the low N rate, which yielded 0.58 Mg ha$^{-1}$ at LCB, while no differences were observed at SCREC in the 2019–2020 winter wheat. In the 2020–2021 winter wheat season, the SCREC location showed higher CP production when a high N rate was applied regardless of timing, at 0.82 Mg ha$^{-1}$ on average, compared to 0.64 Mg ha$^{-1}$ when a low N rate was applied at pre-plant. The LCB location winter wheat CP in 2020–2021 was highest due to the split application yielding 0.6 Mg ha$^{-1}$, followed by a high N pre-plant application yielding 0.47 Mg ha$^{-1}$, and the low N rate pre-plant yielding 0.32 Mg ha$^{-1}$ CP yield.

Summer CP production was influenced by the interaction between winter wheat N application and summer crop species (*p* = 0.0469) and year by summer N application (*p* = 0.0070) at LCB, while the SCREC location was not influenced by any treatments (*p* = 0.1473) (Table 6). The use of a crabgrass monoculture summer forage regardless of winter wheat N application resulted in the lowest summer CP production at an average of 0.16 Mg ha$^{-1}$ compared to all other treatments. The use of cowpeas mixed with crabgrass or monoculture yielded the highest CP regardless of winter wheat N application at an average of 0.43 Mg ha$^{-1}$, except when the mixture followed a split winter wheat N application which yielded 0.34 Mg ha$^{-1}$. No difference in CP yield was observed by summer N application in the 2019 summer season at LCB, with an average CP yield of 0.25 Mg ha$^{-1}$. The 2020 summer CP yield was increased from 0.33 Mg ha$^{-1}$ to 0.47 Mg ha$^{-1}$ by the application of 34 kg N ha$^{-1}$ to the summer forage crops at LCB.

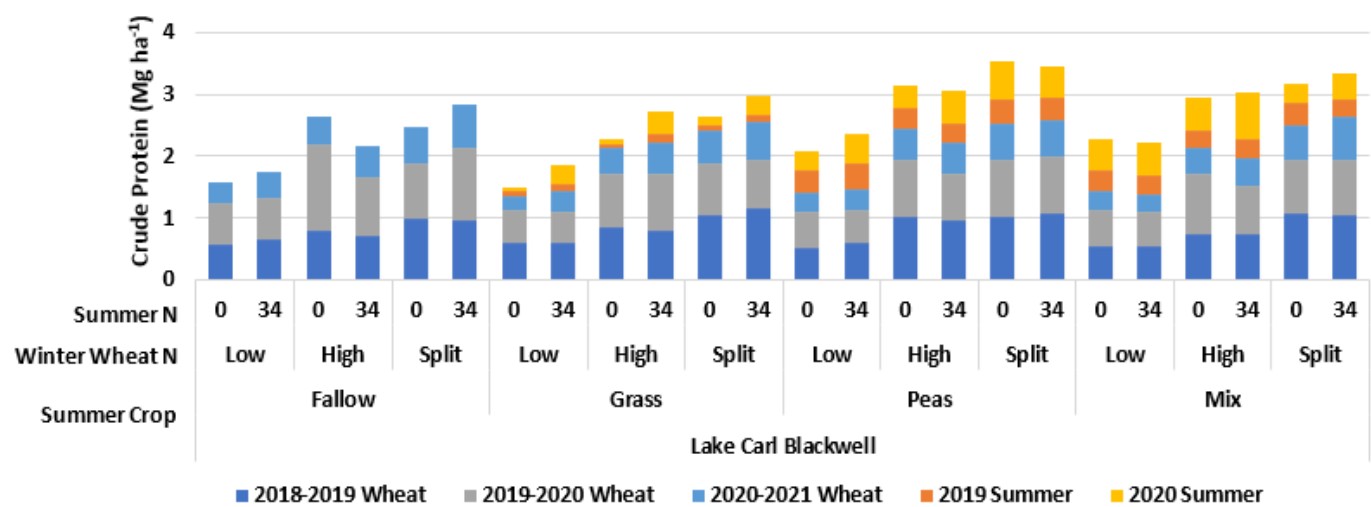

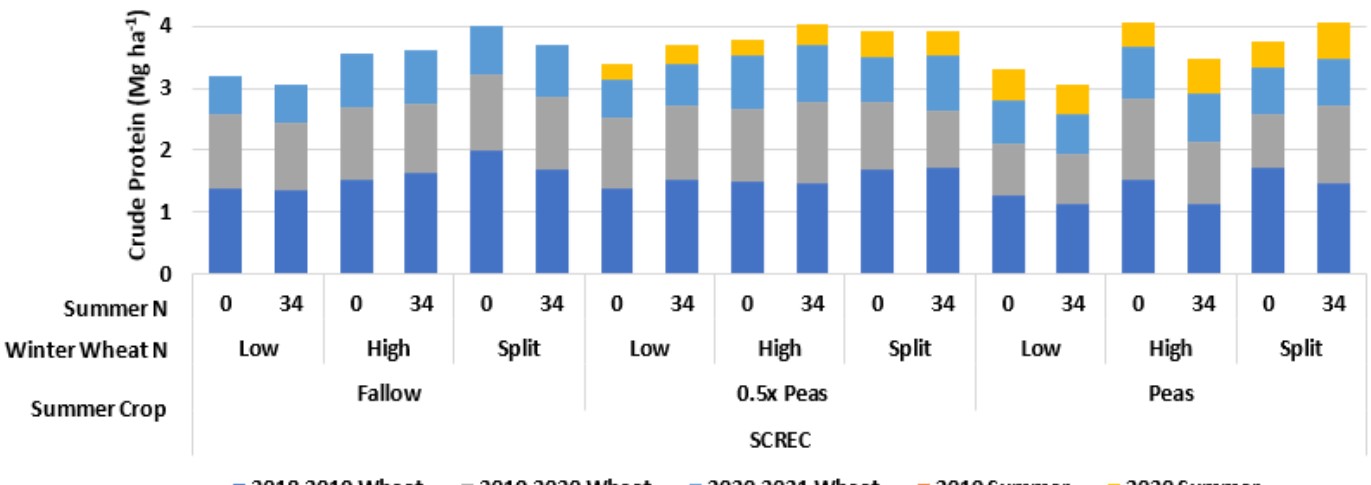

**Figure 2.** Average crude protein yield (Mg ha$^{-1}$) as crude protein multiplied by biomass production from each season of production for LCB (Top) and SCREC (Bottom). Means are presented for each of the summer N application rates (kg N ha$^{-1}$) within each of the summer crop species: fallow, cowpeas (Peas), crabgrass (Grass), cowpea-crabgrass mixture (Mix), and half-planted rate of Cowpeas (0.5× Peas), within each of the winter wheat N applications Low (67 kg N ha$^{-1}$ pre-plant), High (135 kg N ha$^{-1}$ pre-plant), and Split (67 kg N ha$^{-1}$ pre-plant and 67 kg N ha$^{-1}$ top-dress).

**Table 6.** ANOVA table with degrees of freedom (DF), sums of squares, mean squares, *F*-value, and *p*-value for each of the sources of variances for summer crop crude protein yield at two locations in Oklahoma.

| | **Summer Crop Crude Protein Yield** | | | | | | | | | |
| | **SCREC** | | | | | **LCB** | | | | |
| **Source** | **DF** | **Sum of Squares** | **Mean Square** | **F-Value** | ***p*-Value** | **DF** | **Sum of Squares** | **Mean Square** | **F-Value** | ***p*-Value** |
|---|---|---|---|---|---|---|---|---|---|---|
| Model | 14 | 0.50 | 0.04 | 1.60 | 0.1473 | 38 | 4.11 | 0.11 | 5.45 | <0.0001 ** |
| Winter Wheat Nitrogen (WN) | 2 | 0.03 | 0.01 | 0.58 | 0.5661 | 2 | 0.01 | 0.01 | 0.35 | 0.7040 |
| Summer Crop (SC) | 1 | 0.26 | 0.26 | 11.52 | 0.0023 | 2 | 2.02 | 1.01 | 50.91 | <0.0001 ** |
| Summer Nitrogen (SN) | 1 | 0.05 | 0.05 | 2.09 | 0.1608 | 1 | 0.21 | 0.21 | 10.66 | 0.0015 ** |
| Year (YR) | 0 | - | - | - | - | 1 | 0.73 | 0.73 | 36.92 | <0.0001 ** |

**Table 6.** *Cont.*

| | | SCREC | | | | | LCB | | | |
|---|---|---|---|---|---|---|---|---|---|---|
| | | | | | | Summer Crop Crude Protein Yield | | | | |
| Source | DF | Sum of Squares | Mean Square | F-Value | *p*-Value | DF | Sum of Squares | Mean Square | F-Value | *p*-Value |
| WN × SC | 2 | 0.05 | 0.02 | 1.1 | 0.3488 | 4 | 0.20 | 0.05 | 2.50 | 0.0469* |
| WN × SN | 2 | 0.02 | 0.01 | 0.39 | 0.6840 | 2 | 0.07 | 0.03 | 1.70 | 0.1886 |
| SC × SN | 1 | 0.01 | 0.01 | 0.51 | 0.4809 | 2 | 0.09 | 0.05 | 2.30 | 0.1052 |
| YR × WN | 0 | - | - | - | - | 2 | 0.07 | 0.04 | 1.77 | 0.1756 |
| YR × SC | 0 | - | - | - | - | 2 | 0.08 | 0.04 | 2.03 | 0.1367 |
| YR × SN | 0 | - | - | - | - | 1 | 0.15 | 0.15 | 7.57 | 0.0070 ** |
| WN × SC × SN | 2 | 0.03 | 0.01 | 0.60 | 0.5572 | 4 | 0.05 | 0.01 | 0.61 | 0.6537 |
| YR × WN × SC | 0 | - | - | - | - | 4 | 0.19 | 0.05 | 2.34 | 0.0601 |
| YR × WN × SN | 0 | - | - | - | - | 2 | 0.02 | 0.01 | 0.48 | 0.6231 |
| YR × SC × SN | 0 | - | - | - | - | 2 | 0.02 | 0.01 | 0.55 | 0.5814 |
| YR × WN × SC × SN | 0 | - | - | - | - | 4 | 0.05 | 0.01 | 0.57 | 0.6862 |

\* *p*-value significant at 95% level; \*\* *p*-value significant at 99% level.

## 4. Discussion

In this study, environmental variables such as temperature and rainfall were significant factors in location influence, which resulted in locations being analyzed separately. As depicted in Figure 3, the LCB location typically has higher rainfall on a 20-year average than the SCREC location, specifically in the summer months. This trend carried over to the 3-year average of this study, resulting in drier conditions in the summer months near planting of the summer forage crops. These drier conditions at the SCREC reduced the viability of summer forage crops at this location, especially grassy species summer crops that were used in this study.

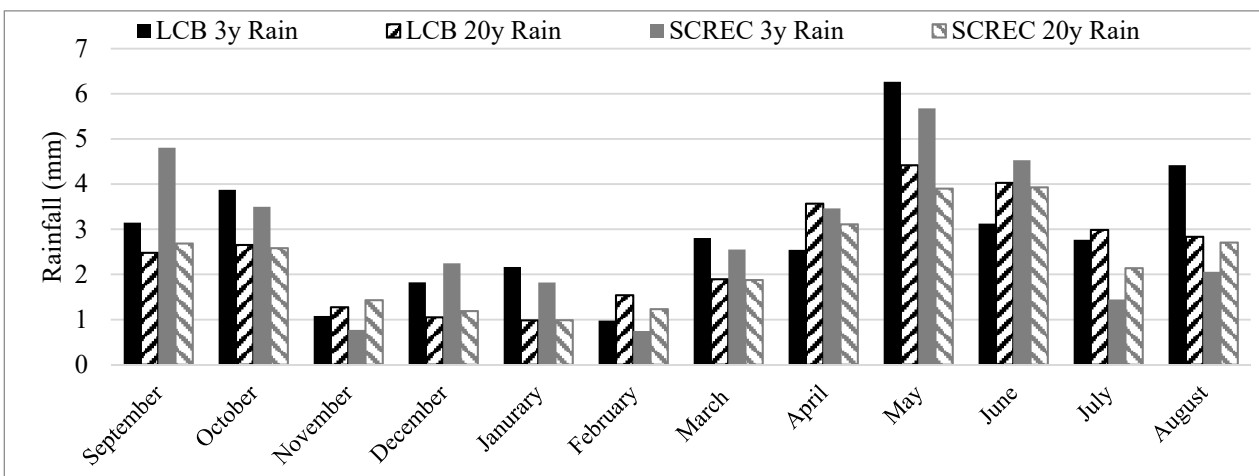

**Figure 3.** Average monthly rainfall amount (mm) for three years of the study (3 y) and 20-year average (20y) at each location. The three-year average starts in Sept. 2018 through May 2021, 20-year average starts in Sept. 2001 through May 2021.

### 4.1. Nitrogen Applications

Winter wheat N applications had an impact on two reported yield factors for wheat and summer crops. The increased application rate of N increased the DM and CP of wheat and summer crops alike. Often, the split application of the low N rate at pre-plant followed by a top-dress application of the low N rate would produce similar or greater DM and CP yield compared to a high N pre-plant application. Increases in winter wheat DM biomass production due to N application have been reported by many previous studies [13–17]. While most of these studies only evaluated the influence of N application rate on a single

pre-plant application, few utilized a split application similar to this study. Naveed et al. [21] reported an increased biomass production when N was applied in a split manner, with 50 or 75% N applied at planting and the remaining applied following the first harvest. The findings of this study are similar, where the split application produced equal or more DM and CP compared to the high pre-plant rate at both locations. The influence of high-rate pre-plant applications on the first harvest and split applications on the second harvests were observed in the individual harvest data. Similarly, Altom et al. [25] reported that in a cereal rye (*Secale cereale*)-wheat-ryegrass (*Lolium multiflorum*) forage system when N was applied in the fall or spring seasons increased DM production of the respective season harvest, however, decreased DM production was observed in the harvest of the season which did not receive N. Further, while split application did not increase total DM compared to when N was applied at a single timing, it did result in a more even harvest across both seasons.

The use of summer N applications only impacted subsequent yields at only one location, LCB, and can be attributed to this research site's lower residual soil N at initiation, as well as the production of summer forage crops in both summer seasons. This shows how the applications of N during the summer would be crucial to the intensification of forage production, making N available for summer crop production as increased N application has been shown to increase DM production of summer forage crops [18,19]. Wiatrak et al. [26] reported in cotton–wheat forage system that even with N applications above the requirement for cotton production, winter wheat forage would still need additional N applications for yield maximization. The summer N application did not influence the DM production of the winter wheat; however, it did drive increases in CP in winter wheat, along with the N applications made to the wheat. This summer's application of N during the summer not only improved the DM and CP production of summer crops but also allowed for more residual N to be utilized for CP synthesis. Similar influences of summer forage crop N application on winter wheat forage CP production were not found by other authors; however, Crusciol and Sarrato [27] found similar results with increased N concentrations in peanut foliage following an N-fertilized pearl millet cover crop.

### 4.2. Cropping Management

Summer forage cropping management had differing results between the two environments in which the study was conducted. Pearl millet, planted as the grassy species, had poor germination at both locations and was outcompeted by invasive crabgrass at LCB. The first summer (2019) at the SCREC location received a long dry period following directly after planting resulting in no stand emergence of summer forage cover crops. Similarly, in the second summer (2020), the grassy species did not germinate due to a similar extended dry period resulting in cowpeas as the only forage cover crop produced at SCREC. The influence of summer crops on winter wheat production was similar at both locations, where the winter wheat production of DM and CP decreased when a summer fallow replacement crop was used. Negative influences of summer forage production on subsequent crops have been reported by previous studies across the Great Plains regions, such as reduced production emphasized during drier production years [3,5]. These decreases in subsequent crop yields can be attributed to decreased plant available water following cover crop production [3,28]. However, other works across the Great Plains show either a positive or neutral response to summer cover crops [28–30]. Holman et al. [28] report no decreases in soil available water at the planting of winter wheat, but no influence was observed on subsequent yield due to cover cropping. This was attributed to adequate time for soil water recharge between summer forage cover cropping and following winter wheat planting [29].

Cowpeas, as the legume species in this study, outyielded the grassy species used, as well as decreased the negative impacts on the subsequent wheat crop. This is similar to Carr et al. [29] and Horn et al. [5], who found subsequent crop yield impacts of summer forage crops to be lessened when a legume was used as opposed to a grassy species forage. The lessened impact could be attributed to the $N_2$ fixation of legumes reducing the requirement of fertilizer N [5]. While summer forages have shown influence on the

subsequent crop in this study as well as others, the influence on total system production was increased regardless of any negative impact on production. In their review of annual forage production impacts on wheat, Carr et al. [29] found increases in production in low N [31] and higher rainfall [3] environments resulting in greater net returns in forage-grain systems. Forage cover crops increase the net returns of cropping systems compared to the use of a fallow period due to increased production and low seed costs [3,28,29]. These returns could be more universal in continuous forage systems, as the production time of the subsequent crop would be less, and with the management of other factors such as N management and species selection, our study, combined with current literature, shows these negative impacts can be mitigated.

## 5. Conclusions

This study found the management of N is necessary for increasing the production of forage systems, especially when continually cropping for forage production. The intensification of the production system by way of summer forage crops was shown to impact the winter wheat with a decrease in winter wheat forage production but increases in CP production. While these subsequent impacts were observed in the wheat production, they were lost in total system production, where the additional forage from the summer crops compensated for the loss of winter wheat forage production.

Our results suggest that the addition of summer forage crops could increase the production of continual winter wheat forage systems in the central Great Plains and that negative impacts on system and winter wheat production can be mitigated by the application of N at a high rate with at least 50% applied at pre-plant. Split applications allow for N to be available at the time of utilization throughout the season, increasing yield and reducing the chance of losses compared to pre-plant applications. Further work would be needed to evaluate the viability of summer forage species for the central Great Plains, as well as evaluate the ideal combination of summer and winter wheat N applications for sustaining intensive forage production in the region.

**Author Contributions:** Conceptualization, B.F. and D.B.A.; methodology, B.F. and D.B.A.; validation, B.F., J.L.B.S., V.R. and D.B.A.; formal analysis, B.F.; investigation, B.F., J.L.B.S., V.R., R.S. and M.S.; resources, D.B.A.; data curation, B.F. and D.B.A.; writing—original draft preparation, B.F. and D.B.A.; writing—review and editing, B.F., J.L.B.S., V.R., R.S. and M.S.; visualization, D.B.A.; supervision, D.B.A.; project administration, B.F.; funding acquisition, D.B.A. All authors have read and agreed to the published version of the manuscript.

**Funding:** This research was funded by Oklahoma Fertilizer Checkoff and the National Institute of Food and Agriculture, U.S. Department of Agriculture, under award number 2019-68012-29888.

**Data Availability Statement:** Not applicable.

**Conflicts of Interest:** The authors declare no conflict of interest.

## Appendix A

**Table A1.** Average dry matter biomass production (Mg ha$^{-1}$) from each season of production for LCB and SCREC.

| Location | Wheat N | Summer Crop | Summer N | 2018–2019 Wheat | 2019 Summer | 2019–2020 Wheat | 2019 Summer | 2020–2021 Wheat | Total Wheat | Total Dry Matter |
|---|---|---|---|---|---|---|---|---|---|---|
| | (kg ha$^{-1}$) | | (kg ha$^{-1}$) | | | Mg ha$^{-1}$ | | | | |
| LCB | 67 | Fallow | 0 | 9.1 | - | 6.5 | - | 3.8 | 19.4 | 19.4 |
| | | | 34 | 10.4 | - | 6.3 | - | 4.0 | 20.7 | 20.7 |
| | | Cowpea | 0 | 8.1 | 2.7 | 5.6 | 2.4 | 3.5 | 17.1 | 22.2 |
| | | | 34 | 9.4 | 3.0 | 5.1 | 4.4 | 3.9 | 18.4 | 25.7 |

**Table A1.** *Cont.*

| Location | Wheat N | Summer Crop | Summer N | 2018–2019 Wheat | 2019 Summer | 2019–2020 Wheat | 2019 Summer | 2020–2021 Wheat | Total Wheat | Total Dry Matter |
|---|---|---|---|---|---|---|---|---|---|---|
| | | Crabgrass | 0 | 9.6 | 1.0 | 5.0 | 0.6 | 2.6 | 17.3 | 18.9 |
| | | | 34 | 9.4 | 1.9 | 5.0 | 3.8 | 4.0 | 17.4 | 23.1 |
| | | Mixture | 0 | 8.3 | 2.4 | 5.6 | 3.8 | 3.6 | 17.5 | 23.8 |
| | | | 34 | 8.7 | 3.0 | 4.9 | 5.5 | 3.4 | 17.1 | 25.6 |
| | 135 | Fallow | 0 | 11.8 | - | 10.5 | - | 5.3 | 27.5 | 27.5 |
| | | | 34 | 10.6 | - | 8.1 | - | 5.4 | 24.2 | 24.2 |
| | | Cowpea | 0 | 14.1 | 2.5 | 7.9 | 2.6 | 5.5 | 27.5 | 32.6 |
| | | | 34 | 12.6 | 3.3 | 6.7 | 4.5 | 5.7 | 25.0 | 32.7 |
| | | Crabgrass | 0 | 12.0 | 0.9 | 7.5 | 1.1 | 4.8 | 24.3 | 26.3 |
| | | | 34 | 11.8 | 2.0 | 8.3 | 4.4 | 5.5 | 25.6 | 32.0 |
| | | Mixture | 0 | 10.8 | 2.2 | 8.7 | 4.2 | 4.7 | 24.2 | 30.6 |
| | | | 34 | 10.8 | 3.0 | 7.3 | 6.3 | 5.0 | 23.1 | 32.4 |
| | 67/67 Split | Fallow | 0 | 14.4 | - | 8.1 | - | 5.3 | 27.8 | 27.8 |
| | | | 34 | 13.7 | - | 9.9 | - | 6.6 | 30.1 | 30.1 |
| | | Cowpea | 0 | 14.9 | 2.9 | 8.5 | 5.2 | 5.4 | 28.8 | 36.3 |
| | | | 34 | 14.6 | 3.2 | 7.8 | 4.8 | 5.2 | 27.7 | 35.7 |
| | | Crabgrass | 0 | 15.8 | 1.0 | 7.7 | 1.5 | 4.8 | 28.3 | 30.8 |
| | | | 34 | 14.9 | 2.0 | 7.2 | 3.3 | 5.0 | 27.1 | 32.4 |
| | | Mixture | 0 | 15.0 | 2.5 | 7.8 | 2.6 | 4.7 | 27.5 | 32.6 |
| | | | 34 | 14.8 | 2.8 | 7.9 | 4.5 | 5.7 | 28.4 | 35.6 |
| SCREC | 67 | Fallow | 0 | 12.6 | - | 9.6 | - | 6.2 | 28.4 | 28.4 |
| | | | 34 | 12.5 | - | 8.6 | - | 6.1 | 27.2 | 27.2 |
| | | Cowpea | 0 | 12.1 | - | 7.6 | 2.7 | 6.6 | 26.3 | 28.3 |
| | | | 34 | 10.8 | - | 7.3 | 2.6 | 6.3 | 24.5 | 26.4 |
| | | 0.5× Cowpea | 0 | 12.3 | - | 9.6 | 1.5 | 6.0 | 28.0 | 29.2 |
| | | | 34 | 13.4 | - | 10.5 | 1.8 | 6.8 | 30.8 | 32.1 |
| | 135 | Fallow | 0 | 13.7 | - | 9.4 | - | 7.7 | 30.9 | 30.9 |
| | | | 34 | 13.5 | - | 8.7 | - | 8.0 | 30.2 | 30.2 |
| | | Cowpea | 0 | 12.1 | - | 11.0 | 2.3 | 8.3 | 31.5 | 33.2 |
| | | | 34 | 10.4 | - | 7.6 | 3.0 | 7.9 | 25.9 | 28.1 |
| | | 0.5× Cowpea | 0 | 12.8 | - | 9.7 | 1.5 | 8.4 | 30.9 | 32.4 |
| | | | 34 | 12.4 | - | 10.3 | 2.0 | 8.7 | 31.4 | 33.4 |
| | 67/67 Split | Fallow | 0 | 17.1 | - | 10.1 | - | 7.0 | 34.2 | 34.2 |
| | | | 34 | 13.5 | - | 10.0 | - | 7.6 | 31.0 | 31.0 |
| | | Cowpea | 0 | 14.4 | - | 7.5 | 2.2 | 6.8 | 28.7 | 30.9 |
| | | | 34 | 12.0 | - | 10.4 | 3.4 | 7.1 | 29.6 | 33.0 |
| | | 0.5× Cowpea | 0 | 14.8 | - | 9.0 | 2.2 | 7.2 | 30.9 | 32.5 |
| | | | 34 | 14.8 | - | 8.4 | 2.3 | 7.2 | 30.4 | 32.2 |

**Table A2.** Average crude protein yield production (Mg ha$^{-1}$) from each season of production for LCB and SCREC.

| Loca-tion | Wheat N | Summer Crop | Summer N | 2018–2019 Wheat | 2019 Summer | 2019–2020 Wheat | 2019 Summer | 2020–2021 Wheat | Total Wheat | Total Dry Matter |
|---|---|---|---|---|---|---|---|---|---|---|
| | (kg ha$^{-1}$) | | (kg ha$^{-1}$) | | | | Mg ha$^{-1}$ | | | |
| LCB | 67 | Fallow | 0 | 0.57 | - | 0.69 | - | 0.32 | 1.57 | 1.57 |
| | | | 34 | 0.66 | - | 0.68 | - | 0.40 | 1.73 | 1.73 |
| | | Cowpea | 0 | 0.51 | 0.38 | 0.60 | 0.31 | 0.29 | 1.41 | 2.09 |
| | | | 34 | 0.60 | 0.41 | 0.55 | 0.49 | 0.33 | 1.48 | 2.37 |
| | | Crabgrass | 0 | 0.61 | 0.07 | 0.51 | 0.06 | 0.24 | 1.35 | 1.48 |
| | | | 34 | 0.59 | 0.12 | 0.51 | 0.31 | 0.34 | 1.35 | 1.79 |
| | | Mixture | 0 | 0.53 | 0.32 | 0.59 | 0.52 | 0.32 | 1.44 | 2.28 |
| | | | 34 | 0.55 | 0.31 | 0.53 | 0.53 | 0.30 | 1.39 | 2.23 |
| | 135 | Fallow | 0 | 0.81 | - | 1.39 | - | 0.45 | 2.65 | 2.65 |
| | | | 34 | 0.70 | - | 0.97 | - | 0.49 | 2.15 | 2.15 |
| | | Cowpea | 0 | 1.03 | 0.33 | 0.92 | 0.36 | 0.51 | 2.46 | 3.15 |
| | | | 34 | 0.95 | 0.31 | 0.78 | 0.52 | 0.49 | 2.22 | 3.05 |
| | | Crabgrass | 0 | 0.84 | 0.07 | 0.87 | 0.09 | 0.42 | 2.13 | 2.28 |
| | | | 34 | 0.79 | 0.14 | 0.93 | 0.38 | 0.50 | 2.22 | 2.74 |
| | | Mixture | 0 | 0.73 | 0.27 | 0.99 | 0.55 | 0.42 | 2.14 | 2.95 |
| | | | 34 | 0.74 | 0.31 | 0.79 | 0.76 | 0.44 | 1.97 | 3.04 |
| | 67/67 Split | Fallow | 0 | 1.00 | - | 0.89 | - | 0.59 | 2.48 | 2.48 |
| | | | 34 | 0.97 | - | 1.18 | - | 0.69 | 2.84 | 2.84 |
| | | Cowpea | 0 | 1.02 | 0.39 | 0.94 | 0.62 | 0.58 | 2.54 | 3.44 |
| | | | 34 | 1.07 | 0.36 | 0.92 | 0.49 | 0.60 | 2.59 | 3.44 |
| | | Crabgrass | 0 | 1.04 | 0.08 | 0.86 | 0.14 | 0.53 | 2.43 | 2.65 |
| | | | 34 | 1.17 | 0.11 | 0.78 | 0.33 | 0.59 | 2.54 | 2.98 |
| | | Mixture | 0 | 1.09 | 0.34 | 0.86 | 0.31 | 0.57 | 2.51 | 3.16 |
| | | | 34 | 1.06 | 0.27 | 0.89 | 0.42 | 0.70 | 2.65 | 3.34 |
| SCREC | 67 | Fallow | 0 | 1.39 | - | 1.19 | - | 0.63 | 3.21 | 3.21 |
| | | | 34 | 1.36 | - | 1.09 | - | 0.61 | 3.07 | 3.07 |
| | | Cowpea | 0 | 1.26 | - | 0.84 | 0.51 | 0.69 | 2.80 | 3.18 |
| | | | 34 | 1.12 | - | 0.83 | 0.49 | 0.64 | 2.59 | 2.95 |
| | | 0.5× Cowpea | 0 | 1.39 | - | 1.15 | 0.26 | 0.60 | 3.14 | 3.33 |
| | | | 34 | 1.54 | - | 1.18 | 0.31 | 0.67 | 3.39 | 3.63 |
| | 135 | Fallow | 0 | 1.53 | - | 1.16 | - | 0.87 | 3.56 | 3.56 |
| | | | 34 | 1.62 | - | 1.12 | - | 0.87 | 3.62 | 3.62 |
| | | Cowpea | 0 | 1.53 | - | 1.31 | 0.40 | 0.83 | 3.67 | 3.98 |
| | | | 34 | 1.12 | - | 1.02 | 0.57 | 0.77 | 2.91 | 3.34 |
| | | 0.5× Cowpea | 0 | 1.49 | - | 1.19 | 0.27 | 0.84 | 3.52 | 3.80 |
| | | | 34 | 1.46 | - | 1.32 | 0.34 | 0.92 | 3.70 | 4.04 |
| | 67/67 Split | Fallow | 0 | 1.98 | - | 1.23 | - | 0.79 | 4.01 | 4.01 |
| | | | 34 | 1.68 | - | 1.18 | - | 0.84 | 3.71 | 3.71 |
| | | Cowpea | 0 | 1.73 | - | 0.85 | 0.42 | 0.75 | 3.33 | 3.75 |
| | | | 34 | 1.45 | - | 1.27 | 0.59 | 0.77 | 3.49 | 4.08 |
| | | 0.5× Cowpea | 0 | 1.68 | - | 1.09 | 0.41 | 0.73 | 3.51 | 3.82 |
| | | | 34 | 1.71 | - | 0.92 | 0.41 | 0.90 | 3.54 | 3.84 |

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
