# Peer review of "Evaluation of Nitrogen and Cropping System Management in Continuous Winter Wheat Forage Production Systems"

_agronomy, doi:10.3390/agronomy13010262_

Round 1

Reviewer 1 Report

The authors propose a manuscript titled “Evaluation of nitrogen and cropping system management in continuous winter wheat forage production systems”.

I suggest the following changes:

References into the text should be changed according to the journal’s instructions

Summary

The introduction in the summary is rather long. Moreover, the summary does not provide enough information about the content of the manuscript

Introduction

I suggest expanding this section, with the addition of more references

Materials and Methods:

It would be important to add information about the varieties included in the study. 
Authors should provide more information about the soils on which the research was conducted, such as the content of carbon, pH, macroelements N, P, K et al.

Results

Mean squares would be useful for readers to be added at the ANOVA tables and the significance with ** or * for 99% and 95% significance, respectively

Discussion

I suggest expanding this section, and discussing results based on references.

References

References should be presented according the journal's instructions

Author Response

REVIEWER 1

I suggest the following changes:

References into the text should be changed according to the journal’s instructions

Response: The authors are grateful for the reviewer feedback. The in-text citations have been updated accordingly.

Summary

The introduction in the summary is rather long. Moreover, the summary does not provide enough information about the content of the manuscript

Response: The summary was reduced according to journal length requirements. Unfortunately, the limitation of 200-word length prevents the addition of more information, while providing adequate background information with insight to objective and results as well.  

Introduction

I suggest expanding this section, with the addition of more references

Response: The authors understand the introduction is short, however the authors took the approach for a concise introduction and while there is a great deal of work focusing on the implementation of cover crops there is a lack of relevant works evaluating nitrogen management within an intensified winter wheat forage production system. The lack of works is one significant reason for the implementation of the study.

Materials and Methods:

It would be important to add information about the varieties included in the study. 
Authors should provide more information about the soils on which the research was conducted, such as the content of carbon, pH, macroelements N, P, K et al.

Response: The variety information has been added for wheat, cowpeas, and pearl millet. Table 2 contains the information on pH, N, P, K, and Total C, and an in-text citation for Table 2 was added.

Results

Mean squares would be useful for readers to be added at the ANOVA tables and the significance with ** or * for 99% and 95% significance, respectively

Response: The authors appreciate the input, however, feel the addition of mean squares to the table would substantially add to the size of the table and the authors feel the addition of * for significance would be redundant with p-values already presented.

Discussion

I suggest expanding this section, and discussing results based on references.

Response: The authors have expanded the discussion of results with greater inclusion of reference-based information.

References

References should be presented according to the journal's instructions

Response: References have been updated to reflect journal requirements

Reviewer 2 Report

The paper by Finch et al. is interesting, timely, and adds useful knowledge to improve the management of forage production systems. The results found can contribute to improve the management of real farms and help farmers in their decision making. The authors have done a lot of useful work, over a considerable period of time (from 2018 to 2021), and in two locations. The paper is well-written and, for all these reasons, I think the work deserves to be publish in Agronomy. However, before publication, I suggest some revisions, as detailed in my attached report. Please, see the attached report.

Author Response

REVIEWER 2

The paper by Finch et al. is interesting, timely, and adds useful knowledge to improve the

management of forage production systems. The results found can contribute to improve the

management of real farms and help farmers in their decision making. The authors have done a

lot of useful work, over a considerable period of time (from 2018 to 2021), and in two locations.

The paper is well-written and, for all these reasons, I think the work deserves to be publish in

Agronomy.

Response: The authors are very thankful for the reviewer’s feedback.

However, before publication, I suggest the following revisions:

- Please, include the species authority when mentioning the scientific names of cowpea

and pearl millet.

Response: Added the species authority for cowpea and pearl millet (Line 95 & 97)

- Please, refer the scientific names including the species authority of crabgrass and wheat, at least the first time it appears in the text.

Response: Added the scientific name and species authority of crabgrass (Line 100) and wheat (line 37)

- I suggest you to divide the section of materials and methods in sub-sections, since it

helps the reader to easily analyze the work (e.g. experimental layout, laboratory

analysis, statistical analysis… or other sub-sections you consider relevant).

Response: Subsections of “field study”, “soil analysis”, “biomass harvest” and, “statistical analysis” were added.

- Table 1 is not cited/mentioned in the text. Please, refer to this table.

Response: Table 1 in-text citation added on line 89

- Soil laboratory analysis are described in lines 108-115, namely regarding

mineral/inorganic nitrogen and soil organic carbon. However, in Table 2, data about

other soil parameters is presented, such as soil pH, P, K, total nitrogen and total carbon.

Please, explain the methods you used to evaluate these relevant soil parameters, as you

did for mineral nitrogen and soil organic carbon.

Response: Total nitrogen was analyzed simultaneously with Total C and added in lines 125 through 126. Analysis methods for pH, P, K, were added in lines 127 through 131.

- In line 110, it is mentioned that soil samples were also collected after the experiment

was complete. If possible, please include this data in your work.

Response: Post season soil analysis was only conducted for nitrogen and carbon measures do to limited resources, and those measures do not address the objectives of this paper. Therefore, the authors have chosen to leave those soil analysis out of this article to maintain concise focus on the production aspects. The soils data pre and post will be in a follow up manuscript focused on the soil chemical, physical results. The authors felt having both yield and soils in a single manuscript would have made for a large and complex paper.

- Table 2 is not cited/mentioned in the text. Please, refer to this table.

Response: Table 2 citation added in the text (line 121)

- In Table 2: i) if possible, please consider specifying the amount of soil nitrate and

ammonium in both locations, instead of generally presenting inorganic N; ii) please

include the amounts of soil organic carbon found in each location, since it was

determined and the Table only presents data for total carbon (if total carbon was not

measured, please correct the column).

Response: The authors understand the interest of specifying the amount of nitrate and ammonium at both locations. Unfortunately, the soil test report from which this data was taken reported the combination of inorganic N. Soil organic carbon was not measured, only total carbon using dry combustion. However given method of analysis and the soil pH, and authors knowledge of the soil, there is a lack of carbonates and therefore TC is highly representative of the TOC.

- Consider standardizing the writing of the p-values. In some parts of the text it is written

in capital letters (e.g. in lines 175-178) and in others in lower case (e.g. in lines 195-198).

Response: Writing of p-values was standardized using lower case “p” throughout the manuscript

- In lines 150 and 211 (Figures 1 and 2, respectively), there is an unnecessary comma after

“cowpeas,”

Response: Thank you for the comment, the unnecessary commas were removed

- Tables A1, A2, and A3 are not cited/mentioned in the text. Please, refer to these tables.

Response: Tables A1 and A2 citations added in text at line 177 & 220. Table A3 was deleted due to data not being presented in manuscript.

- Please, format references according to the journal guidelines.

Response: References were adjusted according to journal guidelines

Round 2

Reviewer 1 Report

Introduction

References into the text should be changed according to the journal’s instructions. The first reference should be reference #1 (instead of #20) and the references should be numbered in consecutive numbers, according to their appearance in the text.

I suggest expanding this section, with the addition of more references

Results

Please provide ANOVA tables with the mean squares and the significance with ** or * for 99% and 95% significance, respectively

Discussion

I suggest expanding this section, and discussing results based on references.
